# OpenReview forum: "All Code, No Thought: Language Models Struggle to Reason in Ciphered Language"
_ICLR.cc/2026/Conference — ICLR 2026 Poster_

### Official Review · Reviewer_p9iy · 2025-10-30

**Soundness:** 3
**Presentation:** 3
**Contribution:** 2
**Rating:** 6
**Confidence:** 3

**Summary:**

This paper investigates whether large language models (LLMs) can perform Chain-of-Thought (CoT) reasoning when the intermediate steps are written in ciphered, translated, or encoded text—a capability termed "ciphered reasoning." This capability is studied as a potential vector for evading AI oversight monitors. Through extensive experiments involving fine-tuning and few-shot prompting across 10 models and 28 different ciphers, the authors identify a notable asymmetry: models maintain high fluency/translation accuracy in ciphered text but suffer a significant drop in mathematical reasoning performance when forced to reason within that ciphered format. The paper correlates this performance drop with the cipher's prevalence in pretraining data and characterizes data/parameter scaling laws, concluding that ciphered reasoning is currently an ineffective evasion tactic for most models.

**Strengths:**

1.  **Systematic and Comprehensive Evaluation:** The study involves an impressive scale of experimentation, testing 10 models against 28 distinct ciphers using rigorous fine-tuning and prompting methods. This systematic approach provides the first large-scale empirical baseline for ciphered reasoning capability in modern LLMs.
2.  **Identification of a Critical Asymmetry:** The core finding—that models can fluently translate or comprehend ciphered text but struggle significantly when executing complex reasoning in that same format—is highly valuable. This asymmetry reveals a fundamental disconnect between the model's linguistic generation capacity and its internal computational mechanism, offering a clear target for future security research.
3.  **Insightful Correlation and Scaling Laws:** The analysis correlating cipher reasoning accuracy with pretraining data prevalence provides a plausible mechanism for the observed behavior. Furthermore, the identified scaling laws offer crucial guidance, suggesting that adversarial fine-tuning for this capability is prohibitively expensive for current frontier models.

**Weaknesses:**

1.  **Lack of Mechanistic Explanation:** The paper clearly states the existence of the asymmetry but stops short of investigating *why* it occurs. It fails to explore the underlying computational or architectural reasons for the collapse of reasoning ability when the surface tokens are altered. This leaves a significant open question regarding the core mechanism driving the reported phenomenon.
2.  **Limited Constructive Exploitation:** Although the paper identifies ciphered reasoning capability as correlating with pretraining data, it primarily utilizes this point for filtering guidance rather than designing a *new* method. For example, it does not propose techniques to intentionally break the correlation for defensive purposes, nor does it design ciphers that optimally exploit the weakness in models, minimizing the ratio of reasoning loss to fluency retention.
3.  **Reliance on Proxy for "Reasoning":** The study heavily relies on mathematical problem-solving accuracy as a proxy for general "reasoning capability." While math is a strong proxy for CoT-as-computation, the conclusions might not fully generalize to non-computational, symbolic, or logical reasoning tasks where the model might behave differently under ciphered conditions.

**Questions:**

Please see the weakness.

---

> ### Author Response · Authors · 2025-11-21
>
> Thank you for your feedback and suggestions. We provide additional experiment results in `Rebuttal.pdf` in the supplemental materials to address your concerns.
>
> > It fails to explore the underlying computational or architectural reasons for the collapse of reasoning ability when the surface tokens are altered. [...]  For example, it does not propose techniques to intentionally break the correlation for defensive purposes, nor does it design ciphers that optimally exploit the weakness in models, minimizing the ratio of reasoning loss to fluency retention.
>
> Thank you for raising this point. Our primary contribution is the first detailed empirical study of CoT reasoning in ciphered text. While ciphered text is a known adversarial attack and potential avenue for CoT monitor evasion, its impact on reasoning has been underexplored. We establish empirical limits on this capability and identify factors that modulate it.
>
> We contribute preliminary evidence towards a mechanistic understanding in Sec. 5.3, where we analyze qualitative cipher attributes that impair reasoning accuracy. We show that ciphers involving syntactic and lexical transformations have a greater impact on models’ reasoning capability than stylistic changes or the addition of distractor text. As noted in our conclusion, both a deeper mechanistic understanding and optimized adversarial cipher design are valuable directions for building on our empirical results.
>
> > While math is a strong proxy for CoT-as-computation, the conclusions might not fully generalize to non-computational, symbolic, or logical reasoning tasks where the model might behave differently under ciphered conditions.
>
> We provide results in Figure 3 of the rebuttal figures for few-shot prompted Sonnet & GPT models on the ZebraLogic \[1\] benchmark, which evaluates LLMs’ ability to solve logic puzzles derived from constraint satisfaction problems. This task does not rely on symbolic reasoning but still requires the model to reason extensively to attain high accuracy.
>
> We find that the magnitude of accuracy drop versus the identity baseline on MATH and ZebraLogic is either similar or the drop on ZebraLogic is much larger, depending on the cipher and model.
>
> \[1\]: https://arxiv.org/abs/2502.01100v1

---

### Official Review · Reviewer_ebcZ · 2025-10-31

**Soundness:** 4
**Presentation:** 3
**Contribution:** 4
**Rating:** 8
**Confidence:** 4

**Summary:**

The authors study capabilities for encoded reasoning in many frontier models, spanning OpenAI, Anthropic, and open-weight models. They study many ciphers, ranging from very simple (adding a space between characters) to advanced ones (like base64). They elicit capabilities via both few-shot prompting and supervised finetuning. They find that, while models can often decode cipher text, doing ciphered reasoning results in substantial drops in accuracy.

**Strengths:**

Originality: Good. While prior work has found that steganography in frontier LLMs appears to be difficult, prior evidence was ad-hoc, and the paper improves substantially on prior works like https://arxiv.org/abs/2507.02737

Quality: Excellent. Authors do extensive experiments to validate their claims and provide insight through follow up analysis.

Clarity: Good. Paper is well written and easy to understand.

Significance: Good. Beyond the object level finding, this work establishes a good playbook for evaluating frontier LLMs on their ciphered reasoning capabilities which could be easily incorporated into a standard evaluation procedure.

**Weaknesses:**

The authors only study instruction-tuned models. For the purpose of 'measuring whether current frontier models can do ciphered reasoning' I think this is fine. However, this is not a very general claim. Also, for the broader scientific purpose of 'determining the relative difficulty of ciphered reasoning vs plaintext reasoning' I think this is an unfair comparison, as mid-training datasets often contain examples of reasoning in plaintext. E.g. there is evidence that Qwen models in particular have been trained extensively on such datasets. https://x.com/stellalisy/status/1927392717593526780?s=46. I would appreciate some experiments on more diverse model families / base models.

The authors mainly consider 'artificial' / pre-existing ciphers; however it is possible that language models can develop their own ciphers via semantic drift, as observed in https://www.antischeming.ai/snippets. This remains an important area for future work.

**Questions:**

Have you tried finetuning via RL? I expect that you would get better ciphered reasoning this way than training purely on SFT examples.

---

> ### Author Response · Authors · 2025-11-21
>
> Thank you for your thoughtful review. We appreciate your recognition of the originality, quality, clarity, and significance of our work. We provide additional experiment results in `Rebuttal.pdf` in the supplemental materials to address your concerns.
>
> > I would appreciate some experiments on more diverse model families / base models.
>
> We additionally evaluate Llama 3.1 8B Instruct and Llama 3.1 8B Base on the `dot between chars`, `rot13 cipher`, `reverse letters in each word`, and `base64` ciphers, following the fine-tuning procedure in Sec. 3.5. Figure 1 of the rebuttal figures shows the results.
>
> We find that the order of magnitude of accuracy drop for each cipher is similar between Llama 3.1 Base, Instruct, and Qwen2.5 7B Instruct. Llama 3.1 Base typically loses slightly less accuracy compared to Instruct models, but the difference is relatively small compared to the magnitude of difference between different ciphers.
>
> > Have you tried finetuning via RL? I expect that you would get better ciphered reasoning this way than training purely on SFT examples.
>
> We provide results for RL via GRPO on Qwen2.5 7B Instruct in Table 1 of the rebuttal figures. We initialize the starting policy from the fine-tuned checkpoint produced by the procedure in Sec. 3.5 and train directly on the MATH train set. See Appendix A.1 in the rebuttal figures for experiment setup details.
>
> When using reinforcement learning, the accuracy is higher than when using supervised fine-tuning, but there remains a large gap between ciphered and unciphered accuracy. Additionally, for the ciphers tested, the ciphers that provide any uplift above the direct answering accuracy are the same when using SFT as when using RL.

---

> > ### Comment · Reviewer_ebcZ · 2025-11-27
> >
> > Thanks, my concerns have been addressed. Have raised my score accordingly.

---

### Official Review · Reviewer_EY1B · 2025-10-31

**Soundness:** 2
**Presentation:** 3
**Contribution:** 3
**Rating:** 6
**Confidence:** 3

**Summary:**

The paper systematically investigates large language models (LLMs) and their ability to reason in ciphered text, introducing the concept of “ciphered reasoning.” The authors evaluate 10 models across 28 different cipher schemes using both fine-tuning and prompting. They find that models can comprehend and accurately translate ciphered text but show a significant drop in reasoning performance, which correlates strongly with the prevalence of each cipher type in the pretraining corpus.

**Strengths:**

S1: The problem definition is novel — this is the first systematic and quantitative study of ciphered chain-of-thought (CoT) reasoning, providing valuable insights for AI safety and monitoring, especially regarding potential CoT-monitoring evasion risks.

S2: The experiments are comprehensive, covering diverse tasks, model types, cipher variations, and multi-scale settings (fine-tuning and few-shot prompting).

**Weaknesses:**

W1: The method itself lacks novelty; it combines existing techniques (fine-tuning, MATH benchmark, BLEU evaluation, cipher generation) with limited theoretical depth.

W2: The evaluation relies solely on MATH 500 as a proxy for reasoning, making the task scope too narrow.

W3: The definition and evaluation criteria for “gaming” behavior are not clearly or rigorously specified.

**Questions:**

The paper reports that billions of tokens are required to approach plain-text reasoning accuracy. Could the authors clarify how sensitive this result is to model initialization and learning rate scheduling? Also, is there evidence that smaller, more targeted datasets could close this gap faster?

---

> ### Author Response · Authors · 2025-11-21
>
> Thank you for your insightful comments. We appreciate that you find our problem definition to be novel and our experiments to be comprehensive. We provide additional experiment results in `Rebuttal.pdf` in the supplemental materials to address your concerns.
>
> > W1: The method itself lacks novelty; it combines existing techniques (fine-tuning, MATH benchmark, BLEU evaluation, cipher generation) with limited theoretical depth.
>
> We agree that our methods leverage existing techniques. As you note, our key contribution is  “the first systematic and quantitative study of ciphered chain-of-thought reasoning… providing valuable insights for AI safety and monitoring”.
>
> > W2: The evaluation relies solely on MATH 500 as a proxy for reasoning, making the task scope too narrow.
>
> We provide results in Figure 3 of the rebuttal figures for few-shot prompted Sonnet & GPT models on the ZebraLogic \[1\] benchmark, which evaluates LLMs’ ability to solve logic puzzles derived from constraint satisfaction problems. This task does not rely on symbolic reasoning but still requires the model to reason extensively to attain high accuracy.
>
> We find that the magnitude of accuracy drop versus the identity baseline on MATH and ZebraLogic is either similar or the drop on ZebraLogic is much larger, depending on the cipher and model.
>
> > W3: The definition and evaluation criteria for “gaming” behavior are not clearly or rigorously specified.
>
> We note that gaming is not the primary focus of this work; we document its occurrence as an example of potentially interesting behavior. However, to clarify what we consider as gaming, we will update the definition to the following in the camera-ready:
>
> Let $\sigma(x_{1:t}, C)$ be a classifier that, given the reasoning trace $x_{1:t}$ of a model, perfectly classifies whether that reasoning trace is ciphered according to the grammar of a cipher $C$. $\sigma$ may not be tractable to compute for many ciphers, so we approximate it with an imperfect classifier $\hat{\sigma}(x_{1:t}, C)$. Our imperfect classifiers described in Sec. 3.1.1 try to capture whether the model has *tried* to follow instructions to cipher its reasoning. Intuitively, we care more about correctly classifying responses that are mostly English as non-adherent than correctly classifying a response that is 1% non-adherent.
>
> *Gaming* refers to responses which are classified as ciphered according to $\hat{\sigma}$, but would not be classified as ciphered according to $\sigma$. In practice, we observe gaming responses cipher stylistic text (such as English words), but keep math expressions and numbers in plain text.
>
> > The paper reports that billions of tokens are required to approach plain-text reasoning accuracy. Could the authors clarify how sensitive this result is to model initialization and learning rate scheduling?
>
> We provide results for the data-scaling experiment described in Sec. 5.2 with Llama 3.1 8B Base & Instruct in Figure 2 of the rebuttal figures. We find that the scaling behavior of Llama 3.1 8B Instruct is similar to that of Qwen2.5 7B Instruct: both fail to reach the same accuracy as the identity model within the 3.7B token train set. Llama 3.1 8B Base can reach the same accuracy as the identity model, but still requires billions of tokens to do so.
>
> Due to compute constraints, we are unable to run a learning rate schedule ablation for the data scaling study. The learning rate schedule used for our experiments was
> picked based on recommendations in the fine-tuning literature \[2\].
>
> \[1\]: https://arxiv.org/abs/2502.01100v1
> \[2\]: https://arxiv.org/abs/2412.13337

---

### Comment · Area_Chair_myrk · 2025-11-25
**Please respond to authors' rebuttal**

Dear Reviewers,

Thanks for your efforts on reviewing this work. Please read the authors' rebuttal and give feedback about whether your concerns have been addressed.

Please note that DONOT fully utilize LLM to directly generate the reviews and responses, ensuring that the reviews are made based on your own expertise, not AI's knowledge. If your review is flagged as LLM generated, it will be reported to PC.

Sincerely,
AC

---

### Meta-Review · Area_Chair_fpsm · 2026-01-07

**Summary:**

Reviewers found the paper to be a strong and timely contribution, highlighting its novelty as the first systematic empirical study of ciphered chain-of-thought reasoning, its extensive experimental scope across models and ciphers, and its relevance to AI safety and monitoring. The main concerns raised were relatively limited in scope, including (i) reliance on mathematical benchmarks as a proxy for reasoning, (ii) limited mechanistic explanation of why reasoning degrades under ciphered formats, and (iii) the initial focus on instruction-tuned models. These issues were satisfactorily addressed in the rebuttal through additional experiments on alternative reasoning benchmarks (e.g., ZebraLogic), evaluations on base models and diverse model families, clarification of definitions (e.g., gaming behavior), and supplementary analyses. Reviewers agreed that the remaining limitations represent natural directions for future work rather than fundamental flaws. Overall, the positive consensus and resolved concerns clearly support acceptance.

**Reviewer Concerns:**

The rebuttal satisfactorily addressed the main technical concerns raised by the reviewers. In particular, the authors expanded beyond MATH as a sole reasoning proxy by adding results on ZebraLogic, evaluated additional model families including base models, and clarified the definition and evaluation of “gaming” behavior. They also provided further evidence on data-scaling behavior and reinforcement learning fine-tuning, which resolved questions about robustness, generality, and training sensitivity. Reviewers explicitly acknowledged that these additions addressed their concerns and updated their scores accordingly.

Some broader limitations remain but do not undermine the paper’s contributions. The work still offers limited mechanistic explanation for why ciphered formats disrupt reasoning, and the exploration of intentionally designed or emergent ciphers is deferred to future work. Additionally, while the added benchmarks improve coverage, reasoning is still primarily evaluated through computational and logic-style tasks. These are reasonable open research directions rather than critical weaknesses, and they do not detract from the paper’s empirical value or suitability for acceptance.

**Reviewer Scores:**

Reviewer EY1B:
This reviewer initially rated the paper as marginally above the acceptance threshold, with concerns about task diversity, clarity of definitions, and training sensitivity. Given that the rebuttal added new benchmarks (ZebraLogic), clarified the definition of gaming behavior, and provided additional scaling and robustness results, I believe this reviewer would likely have modestly increased their score (e.g., from a 6 to a 7), though probably still with some reservations.

Reviewer ebcZ:
This reviewer explicitly stated after the rebuttal that their concerns were addressed and that they raised their score accordingly. Had they participated fully in the discussion from the outset, their final score would clearly remain at a strong accept (8), reflecting high confidence in the paper’s quality and contribution.

Reviewer p9iy:
This reviewer expressed appreciation for the systematic empirical findings but raised higher-level concerns about mechanistic understanding and generalization beyond math reasoning. The rebuttal addressed part of this by adding logic-based benchmarks and clarifying the intended scope of the contribution, but did not fully resolve the mechanistic questions. I believe this reviewer would likely maintain a similar score or increase it slightly (e.g., from 6 to 7), remaining cautiously positive but still noting these as open research directions.

---

### Decision · Program_Chairs · 2026-01-26

Accept (Poster)